# Environmental variation and the evolution of large brains in birds

Ferran Sayol[1], Joan Maspons[1], Oriol Lapiedra[2], Andrew N. Iwaniuk[3], Tamás Székely[4] & Daniel Sol[1,5]

Environmental variability has long been postulated as a major selective force in the evolution of large brains. However, assembling evidence for this hypothesis has proved difficult. Here, by combining brain size information for over 1,200 bird species with remote-sensing analyses to estimate temporal variation in ecosystem productivity, we show that larger brains (relative to body size) are more likely to occur in species exposed to larger environmental variation throughout their geographic range. Our reconstructions of evolutionary trajectories are consistent with the hypothesis that larger brains (relative to body size) evolved when the species invaded more seasonal regions. However, the alternative—that the species already possessed larger brains when they invaded more seasonal regions—cannot be completely ruled out. Regardless of the exact mechanism, our findings provide strong empirical support for the association between large brains and environmental variability.

[1] CREAF, Cerdanyola del Vallès, 08193 Catalonia, Spain. [2] Department of Organismic and Evolutionary Biology, Harvard University, Cambridge, Massachusetts 01238, USA. [3] Department of Neuroscience, Canadian Centre for Behavioural Neuroscience, University of Lethbridge, Lethbridge, Alberta T1K 3M4, Canada. [4] Milner Centre of Evolution, Department of Biology & Biochemistry, University of Bath, Bath BA2 7AY, UK. [5] CSIC, Cerdanyola del Vallès, 08193 Catalonia, Spain. Correspondence and requests for materials should be addressed to F.S. (email: f.sayol@creaf.uab.cat) or to D.S. (email: d.sol@creaf.uab.cat).

Despite wide interest in the evolution of the vertebrate brain, the reasons why some animal lineages—including humans—have evolved disproportionally large brains despite substantial energetic and developmental costs remain contentious. While a variety of selective pressures may have favoured the evolution of enlarged brains[1–3], one that has repeatedly been invoked in the literature is environmental variation. This idea is formally developed in the 'cognitive buffer' hypothesis (CBH, hereafter), which postulates that large brains evolved to facilitate behavioural adjustments to enhance survival under changing conditions[4–6]. Cognition can increase fitness in varying environments by enhancing information gathering and learning, facilitating for instance shifts between different feeding sites or food types to alleviate periods of food scarcity[7–9].

Although the CBH was proposed more than 20 years ago[4], the possibility that environmental variation has shaped brain evolution has garnered only modest empirical support[5,7,10,11]. The absence of firm evidence is striking given the ample support for its main assumption that larger brains (relative to body size) facilitate coping with environmental changes by constructing behavioural responses[12,13]. The current modest support for the CBH hypothesis has led some authors to suggest that the link between brain size and environmental variation could be more complex than often believed[7,14,15]. For example, if growing and maintaining the brain during periods of food scarcity is excessively costly, environmental variability could constrain rather than favour the evolution of large brains[7,14,15]. The complexity of mechanisms linking brain size and environmental variation would explain why attempts to address the CBH in primates have generally been inconclusive[7,10,14,15], despite its relevance to the evolution of large brains and enhanced cognition in humans[16–19]. However, the absence of clear patterns in primates may also reflect that they live mostly in relatively benign tropical environments, where the realized net energy intake experienced by individuals does not necessarily match environmental variability[7].

An excellent group in which to test for a link between brain size and environmental variability is birds, a clade containing species with some of the largest brains, relative to their body size, of any animal[20]. Being among the most widespread land animals, birds experience strikingly different degrees of environmental variation. Moreover, they have been at the forefront of the research into the functional role of enlarged brains in devising behavioural solutions to new challenges[21–23]. Surprisingly, however, only a few studies have addressed the CBH in birds, and the results do not always support it. In parrots, larger brains are associated with higher seasonality in temperature and precipitation[11]. In passerine birds, species that reside the entire year in highly seasonal regions have brains that are substantially larger than those that experience lower environmental variation by migrating to benign areas during the winter[8]. However, phylogenetic reconstructions have revealed that rather than selection for enlarged brains in resident species, the pattern could reflect costs associated with cognitive functions that have become less necessary in migratory species[24].

Here we test whether larger brains are related to environmental variability by means of a phylogenetically-based comparative analysis in birds. We assembled a large database of brain and body size measures of 4,744 individuals of 1,217 species from five continents. We then estimated annual variation in plant productivity (a more direct surrogate of resource variation than temperature and precipitation) throughout their geographic ranges and tested whether species exposed to larger environmental fluctuations within and among years also have relatively larger brains. Having shown this to be the case, we then conducted phylogenetic reconstructions of ancestral traits to ask whether the observed differences are consistent with past selection for enlarged brains in lineages invading regions with high temporal environmental fluctuations. This is achieved by testing whether the evolution of brain size fits better to an adaptive model of phenotypic evolution than a Brownian motion model[25]. Although our main focus was on species residing the entire year in the same region, whose exposure to environmental variation is easier to characterize, we also investigated how species that migrate are affected by environmental variation. In this way, we could reconcile our findings with previous evidence suggesting selection for smaller brains in birds that experience lower degrees of environmental variation by moving to more benign regions during the winter[24].

## Results

**The effect of environmental variation on brain size.** Previous work suggests that selection for larger brains and enhanced learning abilities should be particularly strong in animals inhabiting highly seasonal environments, which demand improved capacity of individuals to track resources that change during the year[9,26]. Consequently, we first asked whether birds exposed to more pronounce seasonal fluctuations in resources are also characterized by disproportionally larger brains. As a way to assess seasonal variation in resource availability, we used remote-sensing analyses to estimate enhanced vegetation indices (EVI) and snow cover within the geographic range of each studied species over a 15-year period[27,28].

In agreement with the CBH, birds residing the whole year in places with higher seasonal variation in EVI have significantly larger brains once accounting for phylogenetic and allometric effects (Table 1). Much of this seasonal variation is associated with latitude. Seasonal changes in EVI are more intense at higher latitudes than at lower latitudes (Supplementary Fig. 1a), reflecting the drop in plant productivity during the cold winters. Supporting the importance of latitude on brain evolution, birds inhabiting regions at higher latitudes tend to have relatively larger brains compared with birds living at lower latitudes (Supplementary Fig. 2, Supplementary Table 1).

At higher latitudes, the period of snow cover is longer than at lower latitudes (Supplementary Fig. 1b). In places with frequent snow, selection for enlarged brains should be particularly strong as food must be obtained under difficult conditions and in a shorter daylight period[9]. Indeed, relative brain size significantly increases with the period of snow cover in resident species (Table 1). High latitude regions do not only experience more snow and higher seasonal variation in plant productivity, but variation in EVI among years is also more pronounced (Supplementary Fig. 1c). This suggests that resources might not only be difficult to track during periods of food scarcity, but they could be unpredictable from year to year[29,30]. Again consistent with theoretical expectations[29,30], brain size (adjusted by body size) is positively associated with variation in EVI among years (Table 1).

**The principal components of environmental variation.** As the above environmental factors are not entirely independent of each other (see Supplementary Table 2), we used a Phylogenetic Principal Component Analysis (PPCA)[31] to produce orthogonal axes describing environmental variation. The first component explains 79% of the variance and has positive loadings on seasonal variation, duration of snow cover and among-year variation (0.97, 0.89 and 0.94 respectively, Supplementary Fig. 1d). This axis therefore represents general environmental variation, with higher values at higher latitudes (Supplementary

**Table 1 | Brain size (dependent variable) in relation to environmental variables in resident birds.**

| Factor | Estimate | SE | t value | Pr ( > \|t\|) |
|---|---|---|---|---|
| *Model 1 (N = 835, P value < 0.001, $R^2$ = 0.87(0.02), λ = 0.92)* | | | | |
| Intercept | −2.53 | 0.12 | −19.86 | <0.001 |
| Log(body size) | 0.59 | 0.01 | 73.24 | <0.001 |
| CV (EVI) within the year | 2.45 | 0.01 | 4.12 | <0.001 |
| *Model 2 (N = 835, P value < 0.001, $R^2$ = 0.86(0.02), λ = 0.92)* | | | | |
| Intercept | −2.50 | 0.13 | −19.51 | <0.001 |
| Log(body size) | 0.59 | 0.01 | 72.47 | <0.001 |
| Log (weeks of snow + 1) | 0.04 | 0.01 | 3.47 | 0.001 |
| *Model 3 (N = 835, P value < 0.001, $R^2$ = 0.87(0.04), λ = 0.90)* | | | | |
| Intercept | −2.61 | 0.13 | −20.44 | <0.001 |
| Log(body size) | 0.59 | 0.01 | 72.60 | <0.001 |
| CV (EVI) among years | 11.18 | 1.97 | 5.67 | <0.001 |
| *Model 4 (N = 835, P value < 0.001, $R^2$ = 0.87(0.04), λ = 0.92)* | | | | |
| Intercept | −2.47 | 0.13 | −19.62 | <0.001 |
| Log(body size) | 0.59 | 0.01 | 73.24 | <0.001 |
| Environmental variation PPC1 | 0.04 | 0.01 | 5.69 | <0.001 |
| Environmental variation PPC2 | 0.02 | 0.01 | 3.04 | 0.002 |

CV, coefficient of variation; EVI, enhanced vegetation index (proxy for resource availability); PPC1 and PPC2, two first axes of a phylogenetic principal component analysis with the environmental variables.
We use a phylogenetic general least square model (PGLS) while controlling for body size and phylogeny. For each model, λ and $R^2$ is shown. In brackets, we also show the $R^2$ of the models once the effect of body size has been removed (see Methods section).

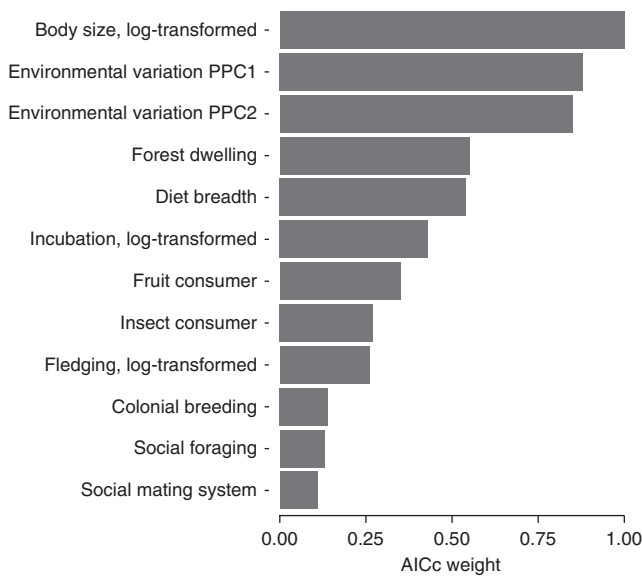

**Figure 1 | Importance of each factor in a model selection approach based on AICc.** A model selection process using PGLS models with Log(Brain size) as a response variable and environmental variation axes (PPC1 and PPC2) and all factors included in the full model (See Supplementary Table 7) as explanatory variables in resident species (N = 242). Here we show the importance of each factor in terms of AICc weights integrated over all possible combinations of models (see Supplementary Table 8 for the best models).

Fig. 1e). The second axis explains 16% of environmental variation and loads positively on variation of EVI among years and negatively on snow cover (component loadings = 0.46, and −0.52, respectively). In contrast with the first axis, this second axis does not describe seasonal variation in EVI (component loading = 0.04) and has higher values at lower latitudes (Supplementary Fig. 1f). Therefore, the second axis primarily reflects variation in EVI among years at lower latitudes (for example, sporadic drought events).

**Additional evidence for the cognitive buffer-hypothesis**. As predicted by the CBH, the two axes describing environmental variation are positively associated with brain size, relative to body size (Table 1), meaning that species that live in more variable environments also tend to have larger brains regardless of the type of variation[29]. This finding is consistent with the view that a relatively larger brain is useful not only in harsh regions (for example, high latitudes with cold winters)[32,33], but also in more benign regions that exhibit substantial year-to-year variation in environmental conditions[11]. Although the model explains over 87% of variation in absolute brain size, only 4% of this variation can be attributed to the environmental variation axes. This limited explanatory power is nonetheless unsurprising considering that body size accounts for the major part of variation and other environmental factors and constraints may also influence brain size evolution[1,34–36]. Yet, it is worth noting that when the analyses were conducted within particular clades the variation in brain size explained by environmental variability is substantial (e.g. ; 19% in Piciformes and 44% in Strigiformes; see Fig. 2). Thus, although the external environment might exert strong selection on cognition and brain size, the evolutionary response is likely to also depend on how the animal interacts with the environment and the extent to which its phenotype constraints certain evolutionary trajectories [29].

**Examining possible confound factors**. The link between relative brain size and the axes of environmental variation (PPC1 and PPC2) in resident birds is not sensitive to phylogenetic uncertainties or potentially confounding variables. First, the results are highly consistent when using 100 randomly selected trees (Supplementary Fig. 3) from the posterior distribution of trees provided by Jetz *et al*[37]. Second, the observed patterns cannot be attributed to changes in body size as being larger or smaller does not co-vary with PPC1 or PPC2 (PGLS: P > 0.12 in all cases; see also refs 38,39). Third, although previous work has suggested that species may be more or less vulnerable to seasonal changes depending on their diet type (for example, frugivory or insectivoury) and preference for buffered habitats (for example,

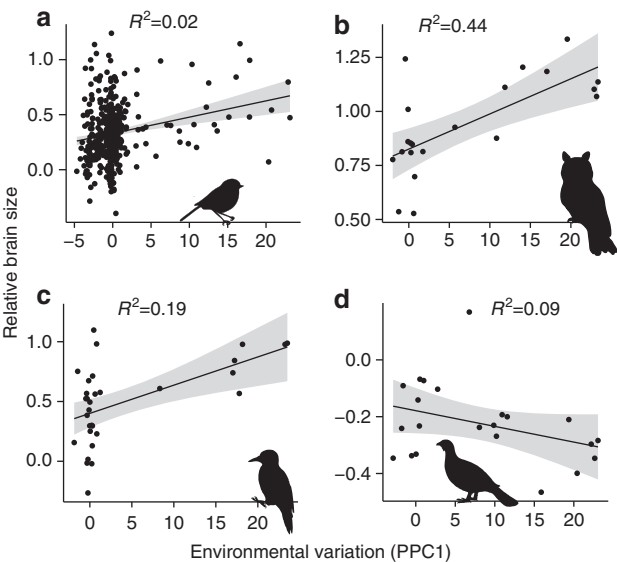

**Figure 2 | Relative brain size and environmental variation (PPC1) within four avian orders.** We tested the effect of environmental variation in four avian orders with representatives in all the latitudinal gradients using PGLS: relative brain size (Mean ± s.e.m.) increase with environmental variation in (**a**) Passeriformes ($0.04 \pm 0.01$, $N = 417$, $P$ value $= 0.01$), (**b**) Strigiformes ($0.07 \pm 0.02$, $N = 21$, $P$ value $= 0.001$) and (**c**) Piciformes ($0.06 \pm 0.02$. $N = 31$, $P$ value $= 0.008$) but not in (**d**) Galliformes ($-0.02 \pm 0.01$, $N = 22$, $P$ value $= 0.097$). The fitted line and the standard error in the figure are derived from the raw data. Silhouette illustrations came from PhyloPic (http://phylopic.org), contributed by various authors under Public domain license.

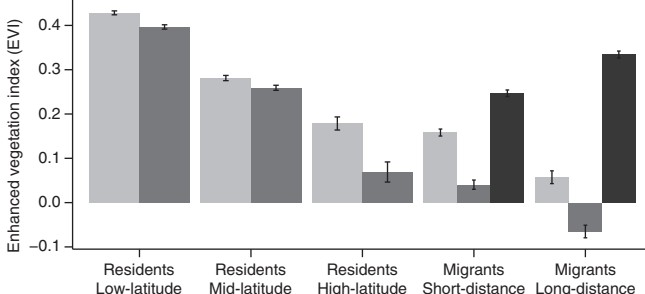

**Figure 3 | Changes in resource availability during the breeding and non-breeding season.** We measured resource availability (mean ± s.e.m.) using the enhanced vegetation index (EVI) in the breeding areas during summer (light grey bars) and winter (dark grey bars). In the breeding areas of residents from higher latitudes ($N = 50$), short-distance ($N = 230$) and long-distance migrants ($N = 87$), there is a larger decrease in EVI during winter (PGLS, $P$ value $< 0.001$, Supplementary Table 10) compared with residents from mid ($N = 326$) and low ($N = 459$) latitudes. Migratory birds skip this decrease in resource availability by moving to wintering areas in lower latitudes (black bars).

forests)[24], including these factors in the models does not alter the conclusions (Supplementary Table 3). Fourth, the association between environmental variation and relative brain size is not indirectly caused by differences in diet generalism (Supplementary Table 4), despite generalists tending to have relatively larger brains and a higher propensity for behavioural innovation[40,41]. Fifth, the effect of environmental variation on relative brain size remains significant even when considering life history traits (particularly developmental periods[2,3,35]) that may constrain brain size evolution (Supplementary Table 5). Sixth, although according to the social intelligence hypothesis the demands of social living might have selected for enlarged brains[1,36], including factors that represent social behaviour (ie, social mating system[1], coloniality[42] and social foraging[36]) does not alter the patterns we report in the present study (Supplementary Table 6). Finally, the axes defining environmental variation are not only significantly associated with brain size even when simultaneously accounting for all the suggested confounds (Supplementary Table 7), but both also consistently appear in all the best models resulting from a model selection procedure (see Supplementary Table 8 for the best models and Fig. 1 for the weight importance of each factor). Although the paucity of information for some traits notably reduced sample size, the model selection and the full model confirmed some previous findings. Thus, larger brains, relative to body size, are also associated with longer incubation periods (see also Supplementary Table 5) and broad diet requirements (Supplementary Table 4).

**The brain-environment association within avian orders.** While the positive association between environmental variation

and brain size holds for the majority of avian orders with representatives in regions with highest environmental variation, a notable exception is Galliformes (Fig. 2). The reasons of such discrepancy are unclear, but could reflect that these species thrive in seasonal regions by means of specialized adaptations rather than plastic behavioural responses. Possible adaptations include a reduced metabolism and specialization on low-quality foods (for example, coniferous needles) that are available the whole year[43]. These specializations would not only constrain the evolution of larger brains, which are energetically costly[44], but also would make exploration and learning less critical for survival[41,45].

**The effect of the environment on migratory birds.** Unlike species that reside the entire year in the same region, migratory birds avoid the drop in resources during the winter by moving to more benign regions (Fig. 3, Supplementary Table 9). Moving also allows them to mitigate variation in productivity across years, as at the wintering areas fluctuations in EVI among years are highly reduced compared with those observed at higher latitudes (Supplementary Fig. 4). Consistent with the lower exposure to environmental variation, in migratory birds the relative size of the brain does not covary with seasonal or among year variation in EVI (Supplementary Table 10). However, the strategy of avoiding the harshest season by moving away is costly, requiring substantial amount of energy to travel between breeding and wintering areas[46]. Interestingly, relative brain size is not only smaller in migrants compared with tropical and temperate residents, in agreement with previous studies[24], but brain size also decreases with travelling distance (Supplementary Table 11). This effect remains robust to the influence of confounding variables (Supplementary Tables 12 and 13, see also Supplementary Fig. 5 for differences within orders). The relationship between migratory distance and relative brain size thus agrees with the energy trade-off hypothesis, which predicts that the brain should be smaller if more energy needs to be allocated to other tissues (for example, pectoral muscle to fly longer distances[34]).

**Evolutionary reconstructions of the evolution of brain size.** To more formally investigate the adaptive nature of the links between

brain size and environmental variation, we used a character stochastic mapping approach to reconstruct transitions across the phylogeny among tropical and temperate regions and, within the latter, between resident and migratory strategies (Fig. 4a). These reconstructions were then used to test whether the subsequent evolutionary trajectories in relative brain size better fit either adaptive (Ornstein–Uhlenbeck—OU) or random (Brownian motion—BM) models of evolution[25,47,48]. Evolutionary reconstructions based on stochastic character mapping[49,50] reveal several independent transitions between tropical and temperate regions and between resident and migratory strategies (Fig. 4b). The best supported evolutionary model is an adaptive model, called OUMV model[51], that assumes the existence of different optima for each selective regime

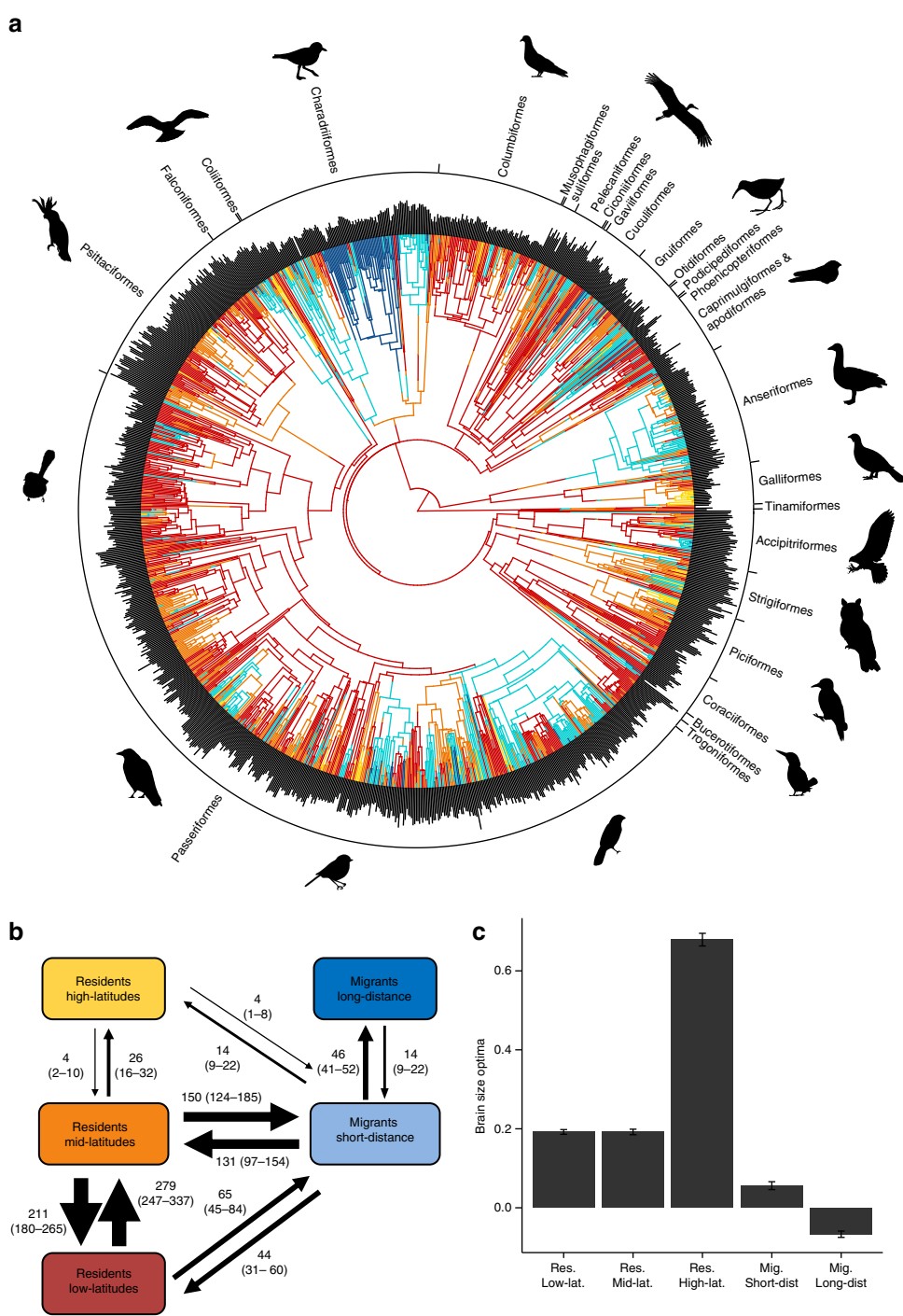

**Figure 4 | Ancestral reconstruction and the evolution of relative brain size.** We reconstructed different character states representing different exposures to environmental variation. An example of a single reconstruction of shifts between migratory behaviours and breeding regions is shown in **a**, where each character state is given a distinct colour (see **b** for colour assignments); outside bars represent the relative brain size of each species, with representative species from the main orders shown. The median number of transitions between different character states and the 97.5 and 2.5% confidence intervals are based on 1,000 reconstructions (**b**). The mean and s.e. of the estimated brain optima under an OUMV model for 100 phylogenies is shown for each category (**c**). Silhouette illustrations came from PhyloPic (http://phylopic.org), contributed by various authors under Public domain license.

and differences in the amount of phenotypic variation ($\sigma^2$) around each optimum (Supplementary Table 14). The estimates of brain size optima from this model are consistent with the hypothesis that species evolved towards larger brains (relative to body size) when moving from tropical to temperate regions (Fig. 4c, Supplementary Table 15). In addition, long distance migrants have a smaller brain optimum compared with residents, further confirming previous evidence[24,33], and exhibit lower variation around this optimum perhaps reflecting a trade-off between brain size and the costs of locomotion[44].

## Discussion

Our results are consistent with the long-held hypothesis that environmental variation may have been an important selective force in the evolution of enlarged brains[4–6]. However, encephalization is a multifaceted process and other selective pressures are likely to have played a role in brain size evolution[1,2,52]. The fact that the explanatory power of environmental variation is low for some avian lineages is indeed consistent with the existence of alternative factors influencing brain size evolution. One factor particularly important according to our analyses is a generalist ecology, which may favour larger brains by frequently exposing individuals to novel or unusual conditions that require behavioural adjustments[40,41,45]. Other factors may also be important despite not being identified in our analyses. For instance, although none of our three measures of sociality was associated with relative brain size, in clear contrast with previous analyses[36,53], this may simply reflect that our metrics do not fully capture key aspects of bird sociality. Finally, our analyses highlight that brain size evolution may also be influenced by constraints[2,3,35], such as the need of a long developmental period to grow a disproportionally larger brain, and selective pressures favouring smaller brains, such as those associated with long-distance movements in migratory birds.

Our approach, however, does not reveal whether species evolved larger brains when they invaded more seasonal regions or instead their ancestors already possessed larger brains when those regions were colonized. Yet given the high metabolic and developmental costs of a large brain[34,54], in highly variable environments the maintenance of a large brain through stabilizing selection seems unlikely unless it provides some sort of benefit that compensates for the costs. If so, the alternative that the ancestors already possessed larger brains when colonized highly variable environments would still be consistent with the CBH. In fact, the two explanations are not mutually exclusive. If a large brain is an important adaptation to cope with environmental variation, highly variable environments should both prevent the establishment of species with small brains and select for larger brains in those that are able to persist there by means of plastic behaviours.

The possibility that a large brain functions, and hence may have evolved, to cope with environmental variation is relevant to understand how animals will be threatened by human-induced rapid environmental changes, notably climate change[55]. If large brains have evolved in slow-lived species to cope with environmental changes, the same species should also be better prepared to cope with climate change and habitat loss. While evidence is accumulating in support of this possibility[21,56,57], it still remains an open question whether adaptive strategies that have been selected in past environments are necessarily appropriate to cope with current environmental challenges[55]. Given the increased concern over how climate change and habitat loss will impact on biodiversity, we anticipate that addressing this issue will be an important avenue for future research.

## Methods

**Brain measurements.** We estimated brain volumes for 4,744 museum specimens from 1,217 species using the endocranial volume technique. This method consists in filling each skull via the foramen magnum with a 50:50 mixture of sizes 10 and 11 lead shot. Once the endocranial cavity was filled, the lead shot was decanted into modified syringes to determine brain volume. This method is highly repeatable and yields a reliable estimate of true brain size[58]. As a way to improve the precision of the measures, we only measured museum specimens of known body size and sex. When more than one specimen per species was examined, we first estimated the average brain volume for each sex and then we estimated the species value as the mean brain volume of the two sexes. Despite the heterogeneous functional organization of the brain, the pallial areas associated with general-domain cognition represent a large fraction of the entire brain, are disproportionally larger in large brained birds and accurately predict variation in the whole brain when allometric effects are appropriately accounted for[59,60].

**Species data collection.** For each species, we also extracted information of the geographical range from BirdLife International (Supplementary Fig. 6 and Supplementary Methods for more details) and used it to (1) assess their breeding latitude (centroid of the breeding range), (2) classify each species as migratory or resident, and (3) estimate migratory distance (difference between breeding and wintering centroids, see Supplementary Fig. 7). In addition, we overlapped each distribution map with layers of the Enhanced Vegetation Index (EVI) over 15 years produced by the Moderate Resolution Imaging Spectroradiometer (MODIS)[27] sensor from NASA. We used the 16 day resolution product from the MODIS sensor to calculate mean and standard deviation values for 23 days of the year. With these data we were able to calculate mean productivity as well as variation among and between years (see Supplementary Methods for details). EVI measures the degree of greenness and provides a proxy to quantify plant productivity at large scales[61,62]. For resident birds, we also obtained data on the persistence of snow cover in winter from MODIS sensor [28]. With the snow cover data, and the CV of EVI among years and along the year, we calculated the First and Second Component of a PCA using the 'phyl.pca' function from 'phytools' R package [63]. Finally we extracted data from the literature for (i) dietary type consumptions, (ii) forest/open habitat type, (iii) incubation and fledging periods, (iv) developmental modes, (v) social mating system and (vi) colonial breeding and (vii) social foraging and complemented the information with the Handbook of the Birds of the World Alive[64]. Diet breadth was then estimated using the frequency of use of distinct dietary types while taking into account that different food items have different degrees of similarity among each other (Supplementary Fig. 8). Snow cover, fledging, incubation and migratory distance were log-transformed and the two PCAs axis were standardized to provide normality. Further details on data acquisition, sources and metric estimations are provided in the Supplementary Methods.

**Phylogenetic hypotheses.** We randomly extracted 100 fully resolved trees from the Bird Tree project[65] for all our species ($n = 1,217$). With the 100 trees, we built the maximum clade credibility tree (summary tree) using TreeAnnotator (a program included in the software BEAST v1.8.0)[66]. Trees from the Bird Tree project include species for which no genetic information is available. Removing the 146 species with no genetic information in our sample does not alter the conclusions.

**Phylogenetic-based approach.** We modelled brain size (log-transformed) as a function of environmental variability and additional covariates by means of Phylogenetic Generalized Least Squares (PGLS) approach[67]. We used the pgls function in the R-package 'caper'[68], which implements GLS models accounting for phylogeny through maximum likelihood estimations of Pagel's λ[69]. We used the consensus phylogenetic tree for all the PGLS analysis, but we re-ran the key analyses with the 100 different trees to account for phylogenetic uncertainty. In birds, enhanced learning abilities are not indicated by absolute brain size, but by the extent to which the brain is either larger or smaller than expected for a given body size[23,59,70]. Consequently, we always included body size (log-transformed and extracted from the same specimens for which brains were measured) as a co-variate when we modeled brain size as a response. However, we also re-ran the analysis with relative brain size, estimated as the residuals of a log-log PGLS of brain against body size (Supplementary Fig. 9), to assess partial $R^2$.

**Phylogenetic reconstructions.** To assess whether historical expansions to more seasonal regions can explain differences in brain size, we used the geographic range of the species to reconstruct transitions between tropical and temperate regions (that is, low-, medium- and high-latitude regions) using stochastic character mapping (SCM)[63]. These latitude categories (see main text and Supplementary Fig. 1) integrate several measures of environmental variation and harshness, such as inter-year and seasonal variation and snow cover. Because migration can reduce environmental variation, evolutionary transitions between residency and migration (short and long distance) were also considered. The combination of these two factors leads to the existence of 5 categories (ie, resident high-latitude, resident medium-latitude, and resident low-latitude, migrant short-

distance and migrant long-distance). Evolutionary transitions among these five selective regimes were reconstructed across a phylogeny encompassing all studied species. This was done using the SCM procedure implemented in the 'simmap' function from R package 'phytools'[63], which estimates the location of evolutionary transitions between categories on a phylogenetic tree. The SCM method allows changes to take place along the tree branches rather than exclusively at the tree nodes[71]. To minimize the potential effects of uncertainty in both tree topologies and phylogenetic reconstructions from the SCM, we used the 100 phylogenies with 10 simulations for each one, resulting in 1,000 phylogenetic trees. To estimate the amount and direction of evolutionary transitions between selective regimes, we used the 'describe.simmap' function over the 1,000 trees and estimated mean and confidence interval for each possible transition (for instance, between Residents medium-latitude and Migrants short-distance).

**Fitting evolutionary models of brain size evolution.** We used the results of the SCM to test if brain size differences between selective regimes are associated with random rates of phenotypic evolution[47] or they are a consequence divergent selective optima for each category[25]. For this purpose, a random set of 100 stochastic character maps were analysed using the R package 'OUwie'[72] to test which evolutionary model best explains the evolution of brain size under the different selective regimes. In this case, we also dealt with allometric effects by estimating the residuals of a log-log PGLS of brain against body size. We considered a variety of Ornstein-Uhlenbeck (OU) models[25] that test for the existence of phenotypical optima (θ) for relative brain size. OU models test the hypothesis that the evolution of a phenotypic trait is non-random, but rather it is the consequence of selective forces pulling this trait to an optimal value that is favoured by natural selection. OU models can either include a single optimum (for example, OU1 model) or consider the possibility that different categories are pulled by natural selection towards different optima. For example, in OUM models smaller brains could be favoured in migratory species while larger brains benefit resident species that experience increased environmental oscillations throughout the year. In the OUMV models, an additional parameter is estimated: the rate of stochastic motion around the optima ($\sigma^2$), representing the amount of brain size variation around the phenotypic optimum estimated for each group. We fitted the following OU models: (1) a simple OU model with a single optimum (θ) and the same α and $\sigma^2$ parameters for all selective regimes ('OU1' model), (2) an 'OUM' model with different optima, and (3) the same OUM model, but with different $\sigma^2$ for each category ('OUMV'). In addition, two Brownian motion (BM) models were also fitted: a single rate 'BM1' model and a 'BMS' model with different rate parameters for each state or phylogeny. Brownian motion models can describe drift, drift-mutation balance and stabilizing selection toward a moving optimum[25]. To assess the most supported model we calculated the Akaike weights for each model based on AICc scores[73]. In addition, we also calculated the Bayesian Information Criterion (BIC), which further penalizes for the inclusion of more parameters. Then, the best evolutionary model was identified from both the AICc and BIC scores and we estimated the mean and the 2.5 and 97.5% confidence intervals for all the parameters. We also tried a more complex model in which another parameter (α) is included as the strength of selection with which natural selection pulls towards a given brain size optimum for each group of species (the so called OUMVA model). However, when using this more complex model, some of the trees gave evolutionary optima that were some orders of magnitude outside the range of existing values of brain size among all bird species and we therefore excluded these models from further analyses.

**Data availability.** All the data generated or analysed during this study are included in Supplementary Data 1.

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

## Acknowledgements

We are grateful to Louis Lefebvre, Simon Reader, Marcos Fernández-Martínez, Mar Unzeta and Simon Ducatez for many helpful suggestions and discussions. This research was supported by funds from the Spanish government (CGL2013-47448-P) to D.S. F.S. was supported by a PhD fellowship FI-DGR 2014 from the Catalan government. O.L. was supported by a Beatriu de Pinós fellowship program. A.N.I. was supported by the Smithsonian Institution Fellowships and Scholarships Program and the Canada Research Chairs program. We thank all the collections and curatorial staff of the museums that allowed us to access their specimens, especially James Dean, Gary Graves, Storrs Olson and Helen James of the National Museum of Natural History (Washington, DC). Geographic information was kindly provided by BirdLife and NatureServe in collaboration with Robert Ridgely, James Zook, Nature Conservancy—Migratory Bird Program, Conservation International—Center for Applied Biodiversity Science, World Wildlife Fund—US, and Environment Canada — WILDSPACE. MODIS 13C1 data were retrieved from the online Data Pool, courtesy of the NASA EOSDIS Land Processes Distributed Active Archive Center (LP DAAC), USGS/Earth Resources Observation and Science (EROS) Center, Sioux Falls, South Dakota, https://lpdaac.usgs.gov/data-set_discovery/modis/modis_products_table/mod13c1.

## Author contributions

F.S. and D.S. conceived and designed the study; F.S., J.M., A.N.I. gathered data; F.S., J.M. and O.L. ran the analyses; F.S. and D.S. wrote the paper; J.M., O.L., A.N.I. and T.S. helped draft the manuscript and approved it.
