## [Peer Review File · Nature Communications]

Reviewers' comments:

Reviewer #1 (Remarks to the Author):

The authors have compiled an extensive dataset of brain size in birds to test the hypothesis that environmental variation drives the evolution of larger brains, as larger brains should presumably determine greater cognitive abilities and possibly increase survival in highly variable environments. They test the hypothesis using various measures of environmental variation and control for an array of potential confounding factors, using phylogenetic comparative approaches. The authors find evidence of associations between larger brains and environmental variation and conclude that more variable environments select for larger brains. Although I find the study interesting, with a large dataset and broadly appropriate phylogenetic comparative methods (but see below), there are several major concerns that undermines the conclusions of the study and especially the claims it makes with regard to directionality and causation. While some of the issues raised below require substantial re-writing, the major concerns involve re-analysis and a far more careful interpretation of results, with regard to causation especially, throughout the whole ms.

Crucially, neither OU models nor stochastic mapping are tests of causation but rather of correlation, and so these methods cannot support any statement that environmental variation 'drives' the evolution of larger brains as claimed in this study. Indeed, stochastic mapping is clearly defined as a correlation method in methodological publications describing it, as it simply quantifies the amount of time 2 variables spend together in a given combination of character states (e.g. see Heulsenbeck et al 2003 *Syst Biol*; Currie & Meade Chapter 10 in 'Modern phylogenetic comparative methods' 2014). OU models, instead, include additional parameters to standard Brownian motion model of evolution which underlies PGLS, a correlational method (see below). As a result, all statements regarding directionality and causation in this ms have to be hugely turned down, starting from the very title. If the authors are instead interested in directionality and order of evolution as evidence consistent with causality, the only comparative method that gets closer to proving this is Pagel's method of correlated evolution for binary traits in a Bayesian framework (see Pagel & Meade 2006 *Am Nat*).

As for OU models used by the authors, these models have come under great criticism in recent years (Ho & Ane' 2013 *The annals of statistics*; Ho & Ane' 2014 *Methods Ecol Evol*; Cooper et al 2016 *Biol J Linn Soc*) and AIC scores cannot adequately discriminate between OU models and its alternative Brownian motion (Boettinger et al 2012 *Evolution*; Cooper et al 2016 *Biol J Linn Soc*). Thus, conclusion based on OU models and model fit based on AIC are highly unreliable. If used, OU models should at least be evaluated under Bayesian framework, which provides a more reliable assessment of the model fit to the data, and even so alpha-values should be interpreted biologically with great caution, particularly when very small as in this study (see Cooper et al 2016 *Biol J Linn Soc*).

The use of multiple trees to account for uncertainty in phylogenetic relationships among species is commendable, although it is possibly not really needed. However, the use of PGLS in maximum likelihood with model averaging employed by the authors is highly

questionable. Model averaging and Akaike weights lead to incorrect conclusions on model fit and the relative importance of variables (e.g. Galipaud et al 2014 *Method Ecol Evol*; Cade 2015 *Ecology*) and this approach should be avoided. Conversely, Bayesian analysis can be easily incorporate multiple trees, provide reliable PGLS parameter estimates and phylogenetic uncertainty on model fit (e.g. De Villemeruil et al 2012 *BMC Evol Biol*). Alternatively, a consensus tree can be used with standard PGLS in maximum likelihood as PGLS is fairly robust to tree misspecification (e.g. Stone 2011 *Syst Biol*). Furthermore, if multiple trees are used a lot more information about the selected trees should be provided (e.g. why choosing a random set of trees rather than the best 100 trees? Why 100, and not, for example 1000? Are the trees from Jetz et al used here among those based on genetic data only, or are they selected from the far more questionable trees incorporating taxonomy for species' without genetic information?).

With regard to the data, it is not clear how most of the independent variables are measured or categorised, and there is a lack of details across the whole ms to fully understand the analyses. For example, it is not clear how diet is classified, i.e. how exactly each category is defined, how the 'similarity' index between dietary categories is computed exactly (similarity based on what specifically?), and why 'diet' is a continuous variable in Table 1 (if it is!) while all supplementary tables present comparison between dietary categories instead. Likewise, why 'snow' is included is not well justified, and why should 'snow' matter but not, for example 'drought'? Surely, birds in more arid environments experience challenging and variable times in finding food and water, as much as, if not more than, temperate birds do when it snows. Why biogeographic regions are also used is a mystery; I don't see how biogeographic regions are of any help in the context of the question asked in this study since each of those regions contains many variable as well as many more constant environments. The same lack of clarity is found with regard to the other variables tested; social monogamy, forest dwelling, sexual size dimorphism, some life history traits randomly included (why including these but not incubation period, egg mass or clutch size?): the justification for testing these variable is weak and poorly explained, and these variables are not well described. As a result, it is unclear what to conclude about the importance of these variables (e.g. diet is significant in Table 1, but not in any supplementary analyses) and so the authors' interpretation of the results (e.g. on diet generalism, page 8).

Finally, the presentation of the analyses is also unclear in places, as some variables are retained and others drop out of models without a clearly identified logic - e.g. why controlling for biogeographic regions for resident birds but not migratory birds? Why CV (EVI) across years, between years and mean EVI are tested for resident birds but only CV (EVI) across years is tested for migrant birds? Why migrant birds are never tested for all other potential confounding factors (social monogamy, forest dwelling, sexual size dimorphism, diet, life history traits etc) but only for EVI? Why is there an analysis within orders for resident but not for migratory birds? Why the environmental variation experienced by migratory birds in their non-breeding range is not considered?

Specific comments

- Why is snow coverage for at least 1 week, not 4 weeks or 3 days, for example?
- Galliformes: an alternative possibility is that these species rely more on fat storage for

coping with unfavourable times than on larger brains and a previous study show fat tissue is negatively associated with brain mass (see Navarrete et al 2011 Nature)

- A model with only body mass is essential to evaluate how much additional variance in brain size environmental factors explain - although the authors state little more (main text end of Page 7), the readers have no means to assess this.

- OU models: if retained, the difference between the alternative OU models must be explained with greater clarity in both the main text and SI - e.g. OUMVA shows up in the main text with no background on it and clear explanation of what this model is and how it differs from the other OU models tested.

- Last page of Discussion: it's unclear how this study reconciles the debate about the evolution of large brain with regard to developmental demands and cognitive benefits and what results in this study support this statement.

- Table S1 is unclear and not enough detail is provided - are these p-values? Correlation coefficients? both p-values and beta estimates or correlation coefficients should be reported

Reviewer #2 (Remarks to the Author):

Environmental variation as a major selective force in the evolution of large brains

In this paper, the authors compare a very large database of bird cranial capacity with environmental variables including EVI, habitat, diet and residence. Overall, I enjoyed this paper and think it is compelling and generally well analysed. The conclusions with regards to EVI and shifts in brain size increase seem robust. Thus, it provides a real contribution to the discussion about the factors driving brain size increase in vertebrates.

I do have a number of queries/suggestions that I feel would further strengthen the paper: these should be addressed prior to publication.

First, given the very large dataset, find it surprising that the authors did not use a model selection approach. Instead, there are a large number of alternative models presented in the SI, but they do not appear to be that systematically reported.

This results in some factors not being considered within the same model. One example of this is the categorical factor of resident-high, resident- mid, resident-low versus migrant-long distance, migrant - short-distance. From what I can tell, these categories are used in the OU analyses for changes in brain size. However, the resident/latitude is only marginally significant in the pglis model it is reported in. Moreover, the OUMVA model optima presented in figure 4c suggest that the effect is driven primarily by a difference in high latitude residents.

Given that snow appears to be an important factor (which could overwhelm the effect of high latitude residents), and that EVI will be highly correlated with both snow cover (by definition snow will occur in places with high EVI) and high latitude, it would be nice (necessary) to use a selection approach to determine whether these three variables are

independently contributing to brain size.

As the results are currently presented, I am left feeling dissatisfied about which of these ecological measures is the most predictive (from the results presented, I feel that most variance can be captured by EVI).

The authors claim that they assess alternative hypotheses including sociality. However, rather than social organisation (which has a strong association with brain size in birds) they evaluate mating system. Not only are these not the same- but in birds mating system is somewhat uninformative as most (~90%) are socially or facultatively monogamous. To test a social hypothesis, they should address variation in flocking structure (note: not group size)- such as pair, bonded groups, aggregations, colony nesters or similar. OR drop this aspect of the analyses- as it is it is a bit disengenuous.

Minor points

Stylistically, I found the slow build up of models and hypotheses a little cumbersome and unnecessary. As I was reading through the manuscript, I wondered why migration wasn't included until fairly late as this seems obvious and essential. I would have preferred to see a global model presented and then effects discussed.

There are a lot of results discussed in the main text that presented in the SI. Some of these results could probably be incorporated into figure legends fairly easily.

Figure 4b is not easy to interpret- nor are the reported results obvious from the data- in fact looking at individual taxon, the effect of latitude is really difficult to pull out. Perhaps having bars for sub-orders or even families for brain size by trait would make the patterns more interpretable?

There is typo in figure 4a.

Reviewer #3 (Remarks to the Author):

I have been asked to review only the remote sensing data aspect of the paper and this is what I have done.

The MODIS EVI data have been used sensibly and appropriately, but the explanations of what and how the data are used needs some work. Specifically:

1. The terminology is not quite correct. For example, "we used data from MODIS VEGETATION INDEX" (supplementary materials page 3). This makes no sense. You have used data from the MODIS sensor which have been processed to provide vegetation indices. One of which is the EVI. Please also check throughout both the SM and main paper that EVI is in capitals etc. To ensure that your terminology is correct please refer to the remote sensing literature. One paper that discusses VI and their use for productivity over time etc is: BOYD, D.S., ALMOND, S., DASH, J., CURRAN, P.J. and HILL, R.A., 2011. Phenology of

vegetation in Southern England from Envisat MERIS terrestrial chlorophyll index (MTCI) data International Journal of Remote Sensing.32(23), 8421-8447

2. In addition to the terminology issues, I had issues with clarity of what had been done. For example, you state that you are using the 16 day product, yet note that "We also calculated the CV of EVI among years, using the global mean of the CV for each day of the year." This is not clear. Equally, what are the 4 measures in "We used the mean EVI from the period December-February (4 measures for 14 years) for non-breeding season and May-July (4 measures for 14 years)"? Both of these examples are from the SI section "Environmental Data".

3. Need to explain why the mean EVI selected? In remote sensing maximum value composites tend to be used, or indeed, the integral of the dataset. Would like some justification.

4. Needs a spell check. For example, "breeding" should be "breeding".

Responses to reviewers

We want to start thanking the three reviewers for all the effort they put into revising our MS. Their comments have really been very useful to improve the clarity and the strength of the study, and we are very grateful for that. Below we address all the reviewer concerns, point by point (in italics).

.....

Reviewer #1 (Remarks to the Author):

The authors have compiled an extensive dataset of brain size in birds to test the hypothesis that environmental variation drives the evolution of larger brains, as larger brains should presumably determine greater cognitive abilities and possibly increase survival in highly variable environments. They test the hypothesis using various measures of environmental variation and control for an array of potential confounding factors, using phylogenetic comparative approaches. The authors find evidence of associations between larger brains and environmental variation and conclude that more variable environments select for larger brains. Although I find the study interesting, with a large dataset and broadly appropriate phylogenetic comparative methods (but see below), there are several major concerns that undermines the conclusions of the study and especially the claims it makes with regard to directionality and causation. While some of the issues raised below require substantial re-writing, the major concerns involve re-analysis and a far more careful interpretation of results, with regard to causation especially, throughout the whole ms.

Crucially, neither OU models nor stochastic mapping are tests of causation but rather of correlation, and so these methods cannot support any statement that environmental variation 'drives' the evolution of larger brains as claimed in this study. Indeed, stochastic mapping is clearly defined as a correlation method in methodological publications describing it, as it simply quantifies the amount of time 2 variables spend together in a given combination of character states (e.g. see Heulsenbeck et al 2003 Syst Biol; Currie & Meade Chapter 10 in 'Modern phylogenetic comparative methods' 2014). OU models, instead, include additional parameters to standard Brownian motion model of evolution which underlies PGLS, a correlational method (see below). As a result, all statements regarding directionality and causation in this ms have to be hugely turned down, starting from the very title.

If the authors are instead interested in directionality and order of evolution as evidence consistent with causality, the only comparative method that gets closer to proving this is Pagel's method of correlated evolution for binary traits in a Bayesian framework (see Pagel & Meade 2006 Am Nat).

Our approach to investigate how the invasion of environmental variable regions has favoured the evolution of disproportionately enlarged brains has been to reconstruct the colonization of these regions in a phylogeny and then examine whether the subsequent changes in brain size better fits either a Brownian or OU model of evolution. This method has been extensively used and, although we agree that it is based correlative evidence, it still provide important insight into the adaptive nature of a trait under a particular selective regime (Beaulieu, 2012 Evolution).

We know well Pagel's method, which we have used in the past (Sol et al. 2010 Plos One). This method might provide cues to causality, although we could not use it because it requires two binary traits and we were dealing with a continuous trait and a variable with five levels. Having said this, in the new version we have removed conclusions regarding causality (starting with the title).

As for OU models used by the authors, these models have come under great criticism in recent years (Ho & Ane' 2013 The annals of statistics; Ho & Ane' 2014 Methods Ecol Evol; Cooper et al 2016 Biol J Linn Soc) (...)

Most of these criticisms are centered on two issues: i) OU models do not work well for small phylogenies and ii) results are largely affected by measurement error. However, our study includes >1000 spp (according to Beaulieu et al in Evolution (2012) the minimum recommended are 150 spp and the methods have the best performance with trees of more than 500spp). In addition, measurement error is negligible in our study because the response variable (brain size) was measured by a single investigator (A. Iwaniuk) and

previous analyses of intraspecific variation and repeatability indicate that the method yields both accurate and precise estimates of brain size (Iwaniuk and Nelson 2002 J. Zool; Franklin et al. 2014 Emu).

(...) and AIC scores cannot adequately discriminate between OU models and its alternative Brownian motion (Boettinger et al 2012 Evolution; Cooper et al 2016 Biol J Linn Soc). Thus, conclusion based on OU models and model fit based on AIC are highly unreliable. If used, OU models should at least be evaluated under Bayesian framework, which provides a more reliable assessment of the model fit to the data, and even so alpha-values should be interpreted biologically with great caution, particularly when very small as in this study (see Cooper et al 2016 Biol J Linn Soc).

To account for this issue, we used AICc instead of AIC because it further penalizes the number of parameters, making BM and OU more comparable. In the revised version, in addition to AICc we use a Bayesian Information Criterion (BIC) to estimate the support for the different evolutionary models. BIC adds higher penalization to the number of parameters in models with big sample size compared to AICc (Johnson & Omland, 2004 Trends Ecol. Evol.). Regarding the alpha values, we agree with the referee that they must be interpreted with caution. In our paper we did not extract conclusions from comparisons of alpha scores among categories. Rather, we were mainly interested in comparing the phenotypic optima among categories. Therefore, for sake of simplicity in the message of the paper, we decided not to include comparisons of alpha scores in the current version of the main MS (although we include the more complex model including categorical the supplementary material).

The use of multiple trees to account for uncertainty in phylogenetic relationships among species is commendable, although it is possibly not really needed. However, the use of PGLS in maximum likelihood with model averaging employed by the authors is highly questionable. Model averaging and Akaike weights lead to incorrect conclusions on model fit and the relative importance of variables (e.g. Galipaud et al 2014 Method Ecol Evol; Cade 2015 Ecology) and this approach should be avoided. Conversely, Bayesian analysis can be easily incorporate multiple trees, provide reliable PGLS parameter estimates and phylogenetic uncertainty on model fit (e.g. De Villemeruil et al 2012 BMC Evol Biol). Alternatively, a consensus tree can be used with standard PGLS in maximum likelihood as PGLS is fairly robust to tree misspecification (e.g. Stone 2011 Syst Biol).

In the PGLS analysis with multiple phylogenies, we do not use a model averaging approach. Instead, we calculated the mean of all the parameters for each of the 100 equally probable trees. We acknowledge this was possibly not clear enough in the previous version of the MS. In the revised version, we use a consensus tree, as suggested by the referee. However, because some authors suggest that using multiple trees is recommended to account for phylogenetic uncertainty (See Garamszegi & Mundry Chapter 12 in 'Modern phylogenetic comparative methods' 2014), in the supplementary material (Figure S3) we also present histograms of the estimates over 100 trees for the key analyses.

Furthermore, if multiple trees are used a lot more information about the selected trees should be provided (e.g. why choosing a random set of trees rather than the best 100 trees? Why 100, and not, for example 1000?)

The decision of using 100 trees is indeed an arbitrary one. However, this number is commonly used in phylogenetic analyses to check the robustness of the results to uncertainty in the phylogeny (Liker et al. 2013 Nat Comm; Gomez-Mestre 2012 Evolution; Dale 2015 Nature; Botero 2014 Molecular Ecology). We choose 100 trees as being a compromise between accounting for phylogenetic uncertainty and doing computationally tractable analysis considering the large number of tips in our phylogeny (N=1217). As we are now using a consensus phylogeny, this issue has nonetheless become less relevant.

Are the trees from Jetz et al used here among those based on genetic data only, or are they selected from the far more questionable trees incorporating taxonomy for species' without genetic information?).

We used both as an attempt to maximize sample size. However, over 88% of our 1217 species have genetic data. This information was missing from the MS and we apology for this. We explain it in the MS methods. We also repeated the key analysis with the higher

quality genetic-based trees and results do not change, although we do not include this in the MS for simplicity.

With regard to the data, it is not clear how most of the independent variables are measured or categorised, and there is a lack of details across the whole ms to fully understand the analyses. For example, it is not clear how diet is classified, i.e. how exactly each category is defined, how the 'similarity' index between dietary categories is computed exactly (similarity based on what specifically?),

In the new version, we have more clearly explained how we defined and categorized each variable (e.g. Diet types; see Methods section). The similarity index was based on nutritional content data. We now provide the raw values of nutritional content we used to calculate the similarity matrix (See Fig. S9) as well as the references used to extract this information (See supplementary methods section).

and why 'diet' is a continuous variable in Table 1 (if it is!) while all supplementary tables present comparison between dietary categories instead.

Diet information is used in two different ways, which we agree could have led to confusion. First, we use two categorical variables (frugivory and insectivory) to describe diets that rely on resources that are highly variable across seasons. We use them as confounding factors to assess whether they could affect the conclusions regarding brain size and environmental variation.

In addition, we use a metric defining diet breadth, that is, the extent to which a species can exploit a variety of food types. Previous studies of brain size in relation to diet have found repeatedly that large brained species tend to be diet generalists.

We used information theory (see De Caceres et al. 2011 Oikos) to estimate a continuous measure of diet breadth based on the frequency use of different diet categories to see if the relationship between environmental variation and brain size can be explained by diet breadth rather than specific diets (i.e., frugivory, insectivory). All this is now explained in the supplementary material.

Likewise, why 'snow' is included is not well justified, and why should 'snow' matter but not, for example 'drought'? Surely, birds in more arid environments experience challenging and variable times in finding food and water, as much as, if not more than, temperate birds do when it snows.

This is another good point. Much of the observed variation in EVI is seasonality related to high latitudes (now we explicitly demonstrate this in the text). In a series of studies, Pravosudov and colleagues have shown that winter 'harshness' significantly affects brain evolution within bird species (see Roth et al. 2010 Proc. Biol. Sci.). For instance, these authors emphasize that harsher habitats are related to cognitive (e.g. learning) and neuroanatomical (e.g. hippocampus) differences according to latitude (and hence snow). We consequently tested whether the presence of snow also helps understand the link between brain size seasonality in evi.

However, we agree that there are other sources of variation, like droughts. Although these effects are included in the variation of EVI (which is the plant greenness and hence will drop in periods of drought), in the revised version we present a phylogenetic PCA to separate different sources of environmental variation. The first axis accounts for most variation and is related to environmental variation at high latitudes. However, the second important axis is related to among-year variation in more tropical regions (e.g. stochastic drought events associated with el Niño). Importantly, brain size is positively associated with the two axes.

Why biogeographic regions are also used is a mystery; I don't see how biogeographic regions are of any help in the context of the question asked in this study since each of those regions contains many variable as well as many more constant environments.

We agree with the reviewer and have now removed biogeographic regions from the analyses.

The same lack of clarity is found with regard to the other variables tested; social monogamy, forest dwelling, sexual size dimorphism, some life history traits randomly included (why including these but not incubation period, egg mass or clutch size?): the justification for testing these variable is weak and poorly explained, and these variables are not well described.

To deal with the possibility that some factors can confound the brain-variation association, we included as covariates those factors that have been proposed to explain brain size evolution (although some of them might have nothing to do with environmental variation, e.g. Social brain hypothesis: Dunbar & Shultz, 2007). The aim of the analysis (previous Table 1) was only to ensure that our results were not biased by other non-environmental factors. Indeed, we note that the fact that environmental variation may affect brain size evolution does not exclude the importance of these other mechanisms. We have further justified the use of confounding factors and have included two additional factors (e.g. incubation period and developmental mode). From all possible life-history factors, we only include developmental traits because they have been shown to be the largest influence on relative brain size variation in birds (Iwaniuk & Nelson, 2003 Can. J. Zool.).

As a result, it is unclear what to conclude about the importance of these variables (e.g. diet is significant in Table 1, but not in any supplementary analyses) and so the authors' interpretation of the results (e.g. on diet generalism, page 8).

These results actually correspond to different variables. One is focused on diet type (frugivory and insectivory) and the other on diet breadth. However, we now have rephrased some of the conclusions about the effect of diet generalism and diet types to be clearer in explaining what factors we are studying as well as more cautious in our interpretation of results (see also our response above).

Finally, the presentation of the analyses is also unclear in places, as some variables are retained and others drop out of models without a clearly identified logic - e.g. why controlling for biogeographic regions for resident birds but not migratory birds? Why CV (EVI) across years, between years and mean EVI are tested for resident birds but only CV (EVI) across years is tested for migrant birds? Why migrant birds are never tested for all other potential confounding factors (social monogamy, forest dwelling, sexual size dimorphism, diet, life history traits etc) but only for EVI?

The novelty of the study was to show that brain size and environmental variation co-vary, and this is the reason why we primarily focused on resident birds (migratory birds reduce variation by moving away). We have added a sentence in the introduction to clarify this. Although the effect of migration on brain size has been investigated in our previous studies (Sol et al. 2010 Plos One), we also included new analyses with the enlarged dataset to deal with the previous suggestion that brain differences between resident and migratory birds were due to selection for smaller brains in migratory species rather than for larger brains in resident species. Our study helps clarify it, providing evidence for both possibilities.

In the case of the confounding factors, we did test them in both resident and migratory birds (Previous Table 1). This was not sufficiently clear in the previous version, however, so we revised this part and tested the confounding factors that might affect brain size (e.g. life-history and social pressures) for both resident and migratory birds separately. We have also included new analyses for migratory birds with EVI mean and EVI across years. We show that these variables are related to brain size in resident birds, but do not affect brain size of migratory birds. Note, however, that we do not have data on snow cover for migratory species and hence we cannot test the PCA axis on migratory birds.

Why is there an analysis within orders for resident but not for migratory birds?

The reason we did not include this analysis was, as explained above, that our focus was on resident birds. We now include a new analysis within orders of migratory birds (Figure S4).

Why the environmental variation experienced by migratory birds in their non-breeding range is not considered?

As we mentioned above, we now include EVI variation within years for migratory species so we now capture the possible environmental variation experienced by migratory birds in both their breeding and non-breeding ranges.

Specific comments

- Why is snow coverage for at least 1 week, not 4 weeks or 3 days, for example?

We used one week to distinguish between regions with frequent snow cover from those that have sporadic snow. To account for potential problems of establishing an arbitrary threshold to classify species between snow dwelling and non-snow dwelling, in this new, revised version we use the weeks of snow cover as a continuous trait. The effect of snow cover on brain size remains significant in these new analyses.

- Galliformes: an alternative possibility is that these species rely more on fat storage for coping with unfavourable times than on larger brains and a previous study show fat tissue is negatively associated with brain mass (see Navarrete et al 2011 Nature)

We concur. Galliformes cope with unfavourable weather by feeding on nutritionally poor food, which results in an expansion of the GI tract. GI tract size can even be altered by changing the amount of fibre that grouse and other species are fed in captivity. This potential trade-off between investment in the GI tract versus the brain agrees with the expensive tissue hypothesis (and hence with Navarrete et al 2011 Nature), so we now include this suggested reference in the revised MS.

- A model with only body mass is essential to evaluate how much additional variance in brain size environmental factors explain - although the authors state little more (main text end of Page 7), the readers have no means to assess this.

We agree with the referee. In the revised version we include the R^2 from models with residual brain size as a response to assess for the variation of brain size explained by environmental variables once the body size effects have been controlled for.

- OU models: if retained, the difference between the alternative OU models must be explained with greater clarity in both the main text and SI - e.g. OUMVA shows up in the main text with no background on it and clear explanation of what this model is and how it differs from the other OU models tested.

We now tried to explain better these models in the MS as well as in the supplementary material. For instance, for simplicity in the message we now leave the more complex model OUMVA for the supplementary material, so we feel that the section on evolutionary models is now easier to follow.

- Last page of Discussion: it's unclear how this study reconciles the debate about the evolution of large brain with regard to developmental demands and cognitive benefits and what results in this study support this statement.

We agree that our idea was not supported sufficiently by the results. In the revised version, we include additional analysis of how developmental periods and developmental modes might influence our conclusions. Despite developmental costs can explain part of brain size variation, environmental variation still significantly affects brain size when including this factors in the models.

- Table S1 is unclear and not enough detail is provided - are these p-values? Correlation coefficients? both p-values and beta estimates or correlation coefficients should be reported

The values provided were p-values. We now include both the estimates and the p-values.

Reviewer #2 (Remarks to the Author):

Environmental variation as a major selective force in the evolution of large brains

In this paper, the authors compare a very large database of bird cranial capacity with environmental variables including EVI, habitat, diet and residence. Overall, I enjoyed this paper and think it is compelling and generally well analysed. The conclusions with regards to EVI and shifts in brain size

increase seem robust. Thus, it provides a real contribution to the discussion about the factors driving brain size increase in vertebrates.

Thank you for the nice comments.

I do have a number of queries/suggestions that I feel would further strengthen the paper: these should be addressed prior to publication.

First, given the very large dataset, find it surprising that the authors did not use a model selection approach. Instead, there are a large number of alternative models presented in the SI, but they do not appear to be that systematically reported. This results in some factors not being considered within the same model.

In the previous version, we included confounding factors that might affect the relationship between brain size and environment. In the revised version, we first test each confounding factor separately to maximize sample size (and we now try to better explain the justification of each confound considered). After that, as suggested by the referee, we do a model selection including all the confounding factors in the same analysis.

One example of this is the categorical factor of resident-high, resident- mid, resident-low versus migrant-long distance, migrant - short-distance. From what I can tell, these categories are used in the OU analyses for changes in brain size. However, the resident/latitude is only marginally significant in the ppls model it is reported in.

We decided to use the latitude categories *instead any of the other environmental variables for three main reasons:*

- i) It is an integrative variable of the variation in the environment (Note that both EVI variation between and among years as well as snow increase in higher latitudes, as shown in fig. S1);*
- ii) The latitude categories have a geographical component, in the sense that we are interested in how the brain changed after the invasion of some regions (e.g. high latitudes) that are highly variable.*
- iii) It is the only variable for which we have information for all the species.*

Moreover, the OUMVA model optima presented in figure 4c suggest that the effect is driven primarily by a difference in high latitude residents.

It is right that the effect is driven primarily by a difference in high latitude residents. However, this is what we expected and agrees with previous evidence that latitudinal differences might affect brain size and learning abilities (Roth et al. 2009 Proc. R. Soc B and 2010 Proc Biol Sci). This issue have been further examined with a phylogenetic PCA, which highlights the importance of latitudinal variation in our results.

Given that snow appears to be an important factor (which could overwhelm the effect of high latitude residents), and that EVI will be highly correlated with both snow cover (by definition snow will occur in places with high EVI) and high latitude, it would be nice (necessary) to use a selection approach to determine whether these three variables are independently contributing to brain size. As the results are currently presented, I am left feeling dissatisfied about which of these ecological measures is the most predictive (from the results presented, I feel that most variance can be captured by EVI).

This is an excellent idea. As the referee points out, the different environmental variables are highly correlated (e.g. $r=0.52$ in the case of variation of EVI within and among years). For this reason we did not included them in the same model. However, it is true that the reader can wonder if both seasonal variation and inter-annual variation are independently correlated with brain size. Or if there are other sources of variation in addition to that related to latitude. In the revised version we addressed this by means of a principal component analysis (PCA) approach. First, we include all the environmental variables (snow, CV of EVI along the year and among years) in a phylogenetic PCA. The three variables mainly load onto the first component in the same direction (87% of the variance), which hence corresponds to intra and among year increased variation of productivity (EVI) and longer periods of snow in higher latitudes. The second PCA axis (9% of the variance) corresponds to increased among-year variation of EVI in tropical regions (with low seasonality and no snow cover). We then use these two orthogonal axes to study how environmental variation can affect brain size. We finally include these two axes and all of the confounding factors in a model

selection approach, as suggested by the referee.

The authors claim that they assess alternative hypotheses including sociality. However, rather than social organisation (which has a strong association with brain size in birds) they evaluate mating system. Not only are these not the same- but in birds mating system is somewhat uninformative as most (~90%) are socially or facultatively monogamous. To test a social hypothesis, they should address variation in flocking structure (note: not group size)- such as pair, bonded groups, aggregations, colony nesters or similar. OR drop this aspect of the analyses- as it is a bit disengenuous.

We agree, although we note that the metrics we used are those used in the most recent tests of the social intelligence hypothesis. In the revised version, we have added information for colonial nesting, which is available for most species. Indeed, more detailed measures of social structure would be desirable, but unfortunately are not available for most species. Addressing these alternatives is not the goal of the study, but we feel that is worth making sure that alternative hypothesis proposed for brain size evolution does not confound our results, so we included analysis with colonial nesting and mating system.

Minor points

Stylistically, I found the slow build up of models and hypotheses a little cumbersome and unnecessary. As I was reading through the manuscript, I wondered why migration wasn't included until fairly late as this seems obvious and essential. I would have preferred to see a global model presented and then effects discussed.

We agree. The main reason why migration was included at the end was that our focus is on resident birds, which are the ones that might be more directly influenced by environmental changes (Indeed, this is what we show in the MS). However, we re-structured the order of the results presented and now we first test the relation between EVI and brain size in resident birds, then in migratory birds and after that we further investigate within order comparisons. The reason why we did not include a global model with all factors was because there is a considerable reduction in sample size when we combine all factors. We still test different confounding factors separately to maximize sample size and after that we do a model selection with all the factors to check the consistency of our conclusions.

There are a lot of results discussed in the main text that presented in the SI. Some of these results could probably be incorporated into figure legends fairly easily.

We agree. We now include some of the models in figures (e.g. Fig. 2) and within the text to avoid so many supplementary tables. Within the SI, we also merged some tables into a single one (e.g. Table S4).

Figure 4b is not easy to interpret- nor are the reported results obvious from the data- in fact looking at individual taxon, the effect of latitude is really difficult to pull out. Perhaps having bars for sub-orders or even families for brain size by trait would make the patterns more interpretable?

We already considered plotting bars for suborders, but the problem is that for some groups, there is a mix of migrant/resident species which would make it difficult to interpret the results in groups with many different categories. Our intention was, to make it easier for the reader to follow the methodology, to show an example of one of the stochastic maps used in the analysis together with the raw data on brain along the phylogeny. Indeed, the important results are provided in the 4a and 4c sections of Figure 4. Provided the above mentioned considerations, we are open to changes to improve the figure.

There is typo in figure 4a.

We fixed it, thanks for spotting this.

Reviewer #3 (Remarks to the Author):

I have been asked to review only the remote sensing data aspect of the paper and this is what I have done.

The MODIS EVI data have been used sensibly and appropriately, but the explanations of what and how the data are used needs some work. Specifically:

1. The terminology is not quite correct. For example, "we used data from MODIS VEGETATION INDEX" (supplementary materials page 3). This makes no sense. You have used data from the MODIS sensor which have been processed to provide vegetation indices. One of which is the EVI.

Good point. We have changed this in the revised version.

Please also check throughout both the SM and main paper that EVI is in capitals etc.

Done

To ensure that your terminology is correct please refer to the remote sensing literature. One paper that discusses EVI and their use for productivity over time etc is: BOYD, D.S., ALMOND, S., DASH, J., CURRAN, P.J. and HILL, R.A., 2011. Phenology of vegetation in Southern England from Envisat MERIS terrestrial chlorophyll index (MTCI) data International Journal of Remote Sensing.32(23), 8421-8447

Thank you for providing that reference. We have included it in the Main ms as well as in the Supplementary Material

2. In addition to the terminology issues, I had issues with clarity of what had been done. For example, you state that you are using the 16 day product, yet note that "We also calculated the CV of EVI among years, using the global mean of the CV for each day of the year." This is not clear.

We acknowledge this might have been a bit confusing in the previous version. Using the 16 day product, we calculate inter-year mean and standard deviation for each of the 23 Julian days provided for the product along the 14 years. With this data, we calculated the CV among years (one for each of the 24 days). We then can use the mean of these values to obtain a single "EVI CV among years". We have tried to better explain this methodology in the revised manuscript.

Equally, what are the 4 measures in "We used the daily mean EVI from the period December-February (4 measures for 14 years) for non-breeding season and May-July (4 measures for 14 years)"? Both of these examples are from the SI section "Environmental Data".

We meant that for the non-breeding period we considered the data from days comprised between December and February (Note that this corresponds to 6 measures, not 4 as we mistakenly reported previously) and days from May to July for breeding period. This corresponds to the Julian days 129, 145, 161, 177, 193 & 209 for the breeding period and the Julian days 337, 353, 1, 17, 33 & 49 for the non-breeding period.

3. Need to explain why the mean EVI selected? In remote sensing maximum value composites tend to be used, or indeed, the integral of the dataset. Would like some justification.

We agree with the reviewer that maximum value composites are preferable as they are a good way to avoid blank lectures (e.g. because of clouds) in the remote sensing data. We indeed were using maximum value composites, as the MOD13C1 (MODIS/Terra Vegetation Indices 16-Day L3 Global 0.05Deg CMG, <http://doi.org/10.5067/MODIS/MOD13C1.006>) product is the result of a maximum value composite of several days of measurement. Using this 16 day product we calculated all the other measures (EVI Annual Mean, EVI CV Among years, EVI CV within the year). We have tried to clarify this in the new version of the SI.

4. Needs a spell check. For example, "breeding" should be "breeding".

Done.

References (in the order of appearance in the responses)

1. Beaulieu, J. M., Jhwueng, D.-C., Boettiger, C. & O'Meara, B. C. Modeling stabilizing selection: expanding the Ornstein-Uhlenbeck model of adaptive evolution. *Evolution* **66**, 2369–83 (2012).
2. Sol, D. *et al.* Evolutionary divergence in brain size between migratory and resident birds. *PLoS One* **5**, e9617 (2010).
3. Iwaniuk, A. N. & Nelson, J. E. Can endocranial volume be used as an estimate of brain size in birds? *J. Zool.* **80**, 16–23 (2002).
4. Franklin, D. C., Garnett, S. T., Luck, G. W., Gutierrez-Ibanez, C. & Iwaniuk, A. N. Relative brain size in Australian birds. *Emu* 160–170 (2014). doi:10.1071/MU13034
5. Johnson, J. B. & Omland, K. S. Model selection in ecology and evolution. *Trends Ecol. Evol.* **19**, 101–108 (2004).
6. Garamszegi, L. Modern Phylogenetic Comparative Methods and Their Application in Evolutionary Biology. (2014). at <<http://link.springer.com/content/pdf/10.1007/978-3-662-43550-2.pdf>>
7. Liker, A., Freckleton, R. P. & Székely, T. The evolution of sex roles in birds is related to adult sex ratio. *Nat. Commun.* **4**, 1587 (2013).
8. Gomez-Mestre, I., Pyron, R. A. & Wiens, J. J. Phylogenetic analyses reveal unexpected patterns in the evolution of reproductive modes in frogs. *Evolution (N. Y.)* **66**, 3687–3700 (2012).
9. Dale, J., Dey, C., Delhey, K., Kempenaers, B. & Valcu, M. The effects of life-history and social selection on male and female plumage coloration. *Nature* **000**, 1–17 (2015).
10. Botero, C. A., Dor, R., McCain, C. M. & Safran, R. J. Environmental harshness is positively correlated with intraspecific divergence in mammals and birds. *Mol. Ecol.* **23**, 259–268 (2014).
11. Cáceres, M. De, Sol, D., Lapedra, O. & Legendre, P. A framework for estimating niche metrics using the resemblance between qualitative resources. *Oikos* 1341–1350 (2011). doi:10.1111/J.1600-0706.2011.19679.x
12. Roth, T. C., LaDage, L. D. & Pravosudov, V. V. Learning capabilities enhanced in harsh environments: a common garden approach. *Proc. Biol. Sci.* **277**, 3187–3193 (2010).
13. Dunbar, R. I. M. & Shultz, S. Evolution in the social brain. *Science* **317**, 1344–7 (2007).
14. Iwaniuk, A. N. & Nelson, J. E. Developmental differences are correlated with relative brain size in birds: a comparative analysis. *Can. J. Zool.* **81**, 1913–1928 (2003).
15. Navarrete, A., van Schaik, C. P. & Isler, K. Energetics and the evolution of human brain size. *Nature* **480**, 91–3 (2011).
16. Roth, T. C. & Pravosudov, V. V. Hippocampal volumes and neuron numbers increase along a gradient of environmental harshness: a large-scale comparison. *Proc. R. Soc. B Biol. Sci.* **276**, 401–405 (2009).
17. Land Processes Distributed Active Archive Center (LP DAAC). MODIS 13C1 Vegetation Indices. Version 5. (2001). doi:<http://dx.doi.org/10.5067/MODIS/MOD13C1.005>

Reviewers' comments:

Reviewer #1 (Remarks to the Author):

The revised version of this ms has substantially improved in the presentation of results, which is now much clearer and easier to follow. There are still some issues of clarity in some areas, which I indicate below. The still outstanding major issue is the interpretation of results with regard to causation (harsh environments 'drive' the evolution of larger brains – e.g. lines 34-5; 214-222) that is not warranted by the analysis given that none of the methods used can detect causation. As I already mentioned in my previous review OU models are just correlational. While the authors acknowledge this is correct in their rebuttal, the ms still presents the results as having demonstrated causation, particularly with regard to OU analyses. Because OU is correlational, it will provide the same results whether larger brains evolve as a consequence of selection in harsh environments OR whether larger brained birds moved into harsh environments. What the OU results of the study show is simply that there are different optima for relative brain size across the 5 categories of birds tested (low/mid/high latitude and short/long distant migrant) but OU models are blind to how such associations evolved, i.e. whether larger brains evolve in response to harsh environments (as the authors believe), or whether birds with larger brains are pre-adapted to survive in harsher environments, in which they moved *after* having evolved larger brains. In fact one of the main problems with OU models is that results are interpreted by different people as supporting very different processes (e.g. niche conservatism, stabilizing selection) despite OU models do not test for any specific processes and cannot discriminate between alternative processes (see Cooper et al 2016 Biol J Lin Soc). Stabilizing selection, for example, could explain the results of this study e.g. larger brains evolve first in non-harsh environments and selection maintains an advantage of larger brains in birds that subsequently move into harsh habitats.

Overall, the ms present strong evidence of a correlation between large brains and highly seasonal, temporally variable environments at higher latitude; these results are however consistent with both the hypothesis that larger brains evolve in response to selection in harsh environment and the hypothesis that larger brains evolve first and are a pre-adaptation subsequently advantageous to species colonizing next harsh environments. This must be made crystal clear in the discussion and interpretation of results.

Specific comments

- Lines 86: why 'disproportionally'? do you mean relative to body size?
- Lines 128-136: even if the authors also use PCA, a full model with all independent variables should be presented (at least body size, EVI within and across years, weeks of snow). In addition, the authors state in their response that the variables are highly collinear ($r \sim 0.5$) but collinearity is generally considered high when $r > 0.7$. Furthermore, the authors present no statistical test showing the extent of collinearity between predictors – Variance Inflation Factors are ideal in this regard, or at least a correlation matrix between predictors should be included.
- Line 140: but El Nino has global effects, including milder winters at higher latitude.
- Lines 153-170: the presentation of these results is now much easier to follow. However, it is still not clear *how* 'social pressures' (lines 166-8) should affect brain size; here you

present analyses on coloniality and pair-bonding (table S5) but you left the reader wondering in what way these factors should matter for brain evolution. Specific predictions for these factors should be briefly mentioned in the main text as you have successfully done for the other potential confounding factors.

- Lines 272: how were phylogenetic relationships 'inferred when genetic data is not available'?
- Line 274-275: it is not clear if these 146 species do not have genetic data but are still in Jetz et al's trees, or if they are not at all in any available tree and the authors somehow managed to incorporate them into Jetz et al's tree.
- pPCA: loadings on PC axes and % variance must be given in a table to help readers evaluate if the interpretation of what the PC axes corresponds to what the authors state – Fig. S1d is very unclear (with – apparently species names plotted on it?!) and does not help much clarifying this.
- Fig.2: information about how the fit line on these graphs is derived is missing.
- Fig 3 title: the ancestral reconstruction is not about 'selective regimes' but on the habits of the species tested.
- Fig. 3b: confidence intervals of the transitions presented in this figure should be provided (e.g. in a Table in the SI).
- Fig 3c: this seems to match the less reliable OUMVA (Table S14) rather than OUMV (Table S13) as stated in the figure legend.
- FigS1 b and c are presented in reversed order in the text relative to the figure.
- SI line 40-42: it is not clear why 6,000Km is used to split migrants into short- and long-distant migrants since the 'pit' seems to be around 5000Km.
- SI Lines 133-4: diet categories and forest-dwelling are poorly defined. Do you consider a bird as consuming fruit if it occasionally, regularly or only eats fruit? Same for insects. How did you handle within-species variation, e.g. due to different populations having different diet preferences? Do forest dwellers include birds that exclusively live in forests of any kind or that make at least some regular use of forests? Do you consider all forests equivalent?
- SI lines 148-150: where does the information for % water, lipid etc. come from for each type of food and how representative is it for all the great diversity in food sources across such a large scale? Ref. 22 is restricted to very few seed types.
- SI lines 156-65: clear definitions of each category of development and social mating system must be given. Across text, SI, and tables the wording used for each category and variable keep changing – please be consistent to help readers find the information more easily (e.g. social mating system in the text of the SI becomes pair-bonding in the tables).
- SI line 191-5: CI for each possible transition must be reported
- Figure S8: phylogenetic 'correlation' is inaccurate definition here – I suppose you mean 'regression' from which you computed residuals
- Fig S9a: I suppose these are percentages; please clarify. If percentages, 'carrion' does not sum up to 100.
- Table S6: What are the numbers for each variable in this table? Beta estimates? P-values? T-values? Why does mating system have no numbers? How are the df computed?
- Table S7: how is the phylogenetic anova carried out? Do you mean a PGLS model with a discrete predictor variable or Garland's phylogenetic ANOVA using randomization? How can (b) represent p-values? Also for (b) there is an extra number for Resident high lat.
- Tables S13-14: unclear what the last 2 columns in this table report.

Reviewer #2 (Remarks to the Author):

I appreciate that the authors have taken the time to consider and address the reviewers comments from the previous draft. I still think this paper represents an important contribution. However, the model selection approach is still hugely ungainly and difficult to interpret. I cannot understand why, with such a large sample size, they cannot do a global model and evaluate the relative contribution of each factor. As it is, there is a plethora of different tables and models and approaches (i.e. IC versus hypothesis testing). I am less convinced by the results than I was in the previous version. There are clearly ecological and behavioural variables that are associated with brain size, but the current version does not make it straightforward to assess relative contribution.

The addition of PCA is reasonable, but that there is no difference (and very little variation between low and mid-latitude environments (figure S1e) makes me question just exactly what these PCs are measuring (there are highly seasonal temperate and tropical environments, but this does not appear to be captured by these data).

I would like the authors to revisit the presentation of their results to make the alternative models easier to follow.

Also, I do think there needs to be a more explicit discussion of causality and what we can infer from these results (or not)- as it is not possible to determine easily whether large brained lineages can colonise northern climates or whether large brain size is selected for in high latitudes.

Our answers to reviewers are presented in italics.

Reviewers' comments:

Reviewer #1 (Remarks to the Author):

The revised version of this ms has substantially improved in the presentation of results, which is now much clearer and easier to follow.

We sincerely appreciate the reviewer's effort to help us improve the MS.

There are still some issues of clarity in some areas, which I indicate below. The still outstanding major issue is the interpretation of results with regard to causation (harsh environments 'drive' the evolution of larger brains –e.g. lines 34-5; 214-222) that is not warranted by the analysis given that none of the methods used can detect causation. As I already mentioned in my previous review OU models are just correlational. While the authors acknowledge this is correct in their rebuttal, the ms still presents the results as having demonstrated causation, particularly with regard to OU analyses. Because OU is correlational, it will provide the same results whether larger brains evolve as a consequence of selection in harsh environments OR whether larger brained birds moved into harsh environments. What the OU results of the study show is simply that there are different optima for relative brain size across the 5 categories of birds tested (low/mid/high latitude and short/long distant migrant) but OU models are blind to how such associations evolved, i.e. whether larger brains evolve in response to harsh environments (as the authors believe), or whether birds with larger brains are pre-adapted to survive in harsher environments, in which they moved *after* having evolved larger brains. In fact one of the main problems with OU models is that results are interpreted by different people as supporting very different processes (e.g. niche conservatism, stabilizing selection) despite OU models do not test for any specific processes and cannot discriminate between alternative processes (see Cooper et al 2016 Biol J Lin Soc). Stabilizing selection, for example, could explain the results of this study e.g. larger brains evolve first in non-harsh environments and selection maintains an advantage of larger brains in birds that subsequently move into harsh habitats. Overall, the ms present strong evidence of a correlation between large brains and highly seasonal, temporally variable environments at higher latitude; these results are however consistent with both the hypothesis that larger brains evolve in response to selection in harsh environment and the hypothesis that larger brains evolve first and are a pre-adaptation subsequently advantageous to species colonizing next harsh environments. This must be made crystal clear in the discussion and interpretation of results.

We agree with the reviewers' opinion and apologize if our latest version was not clear enough. We have tried to further clarify this in the new version. Specifically, we include a section in the discussion explicitly stating that (lines 242-244) "our approach does not reveal whether species evolved larger brains when they invaded more seasonal regions or instead their ancestors already possessed larger brains when those regions were colonized." However, we also note that given the high metabolic and developmental costs of large brain, "the maintenance of large brains through stabilizing selection seems unlikely unless it provides some sort of benefit that

compensate the costs". Therefore, the possibility that the ancestors already possessed larger brains when those regions were colonized would still be consistent with the CBH. In addition, we also note that "If a large brain is an important adaptation to cope with environmental variation, highly variable environments should both prevent the establishment of species with small brains and select for larger brains in those that are able to persist there by means of plastic behaviours." Finally, we have revised the entire text to avoid statements that can be misinterpreted in terms of causation (e.g. "drive").

Specific comments

- Lines 86: why 'disproportionally'? do you mean relative to body size?

Yes, we meant relative to body size. Throughout the text, we now use "relative to body size" instead of "disproportional" to avoid confusions.

- Lines 128-136: even if the authors also use PCA, a full model with all independent variables should be presented (at least body size, EVI within and across years, weeks of snow). In addition, the authors state in their response that the variables are highly collinear ($r \sim 0.5$) but collinearity is generally considered high when $r > 0.7$. Furthermore, the authors present no statistical test showing the extent of collinearity between predictors – Variance Inflation Factors are ideal in this regard, or at least a correlation matrix between predictors should be included.

We agree that a full model would be interesting, but it would not be correct because the environmental variables are collinear (the VIF is 6.6 and it is considered to be problematic when > 2). When this happens, it is recommended to combine the predictors in a PCA (e.g. Freckleton 2011 Behav Ecol Sociobiol), which will be independent and can be used in the same model, so this was our approach. Now we more clearly explain this in the methods and we include the correlation matrix of the environmental variables as a supplementary table (Table S2).

- Line 140: but El Niño has global effects, including milder winters at higher latitude.

We removed any reference to "El Niño" events, and instead talk about recurrent droughts in tropical and subtropical environments.

- Lines 153-170: the presentation of these results is now much easier to follow. However, it is still not clear *how* 'social pressures' (lines 166-8) should affect brain size; here you present analyses on coloniality and pair-bonding (table S5) but you left the reader wondering in what way these factors should matter for brain evolution. Specific predictions for these factors should be briefly mentioned in the main text as you have successfully done for the other potential confounding factors.

We now have added a phrase describing the logic of the social intelligence hypothesis (lines 169-174): "although according to the social intelligence hypothesis the demands of social living might have selected for enlarged brains^{1,36}, including factors that represent social behaviour (i.e. social mating system¹ and coloniality⁴³) does not alter the patterns we report in the present study (Supplementary Table 6); indeed, our analyses do not provide any evidence that species that are socially monogamous and/or that breed in colonies have larger brains".

- Lines 272: how were phylogenetic relationships 'inferred when genetic data is not available'?

We did not infer any relationships among species in the phylogenetic trees we used. Instead, we used complete trees taken from Jetz 2012 (see below).

- Line 274-275: it is not clear if these 146 species do not have genetic data but are still in Jetz et al's trees, or if they are not at all in any available tree and the authors somehow managed to incorporate them into Jetz et al's tree.

We took complete phylogenies from Jetz et al. 2012 (which include both species with genetic and no genetic data). We have clarified this in the text (lines 297-301). We note that removing the species with no genetic data from the analysis do not alter the conclusions of the study.

- pPCA: loadings on PC axes and % variance must be given in a table to help readers evaluate if the interpretation of what the PC axes corresponds to what the authors state

Although we already provided in the text the % variance for PC1 and PC2 and the loading of the PC1, we now have also added the loadings for PC2. As now all the information is presented in the text (lines 128-136), we feel the table unnecessary. We have also added a clearer interpretation of each axis in the same paragraph in the revised MS.

- Fig. S1d is very unclear (with – apparently species names plotted on it?!) and does not help much clarifying this.

In FigS1d, we removed species names to make the figures clearer.

- Fig.2: information about how the fit line on these graphs is derived is missing.

True, we added this information.

- Fig 3 title: the ancestral reconstruction is not about 'selective regimes' but on the habits of the species tested.

We have eliminated "selective regimes", as suggested by the reviewer.

- Fig. 3b: confidence intervals of the transitions presented in this figure should be provided (e.g. in a Table in the SI).

We now provide this information (inside the Fig. 3b).

- Fig 3c: this seems to match the less reliable OUMVA (Table S14) rather than OUMV (Table S13) as stated in the figure legend.

Good point. In the revised version we included the new estimates for the OUMV in the SI but we mistakenly attached the older version of figure 3c. We now include the correct figure and thank the reviewer for noticing the mistake.

- FigS1 b and c are presented in reversed order in the text relative to the figure.

We fixed it.

- SI line 40-42: it is not clear why 6,000Km is used to split migrants into short- and long-distant migrants since the 'pit' seems to be around 5000Km.

This was a typing error in the MS, because we indeed did the split in 5000 km. We fixed it.

- SI Lines 133-4: diet categories and forest-dwelling are poorly defined. Do you consider a bird as consuming fruit if it occasionally, regularly or only eats fruit? Same for insects. How did you handle within-species variation, e.g. due to different populations having different diet preferences? Do forest dwellers include birds that exclusively live in forests of any kind or that make at least some regular use of forests? Do you consider all forests equivalent?

We consider a frugivorous or insectivorous species that regularly (or only) eat the specific food source but not if they consume it occasionally. We have substantially expanded the information on the supplementary methods to explain how we distinguish frequently consumed from rarely consumed food (lines 108-115 in SI). About the variation inside species, if available, we averaged the information of the populations into the species level (If one population only eats fruit but another population of the same species eats fruit and insects, we consider the whole species to eat both fruit and insects). However, detailed information of different populations is not so common and in the majority of cases the main source of information was at the species level (SI lines 115-120). Forest dwellers include birds that regularly use forests (we do not consider species using forests occasionally) and we considered all types of forests or woodlands above 3 meters (e.g. shrublands or bushlands were not considered forests). We also included this information in the supplementary method section (SI lines 142-144).

- SI lines 148-150: where does the information for % water, lipid etc. come from for each type of food and how representative is it for all the great diversity in food sources across such a large scale? Ref. 22 is restricted to very few seed types.

We use values estimated from representatives of each food type available in the literature. Although the sources used might be a limited representative of each food type, we think that this is still a more correct approach than to calculate diet breadths assuming that all the food type are equivalent. This is explained in detail in one of our previous papers (De Cáceres et al. 2011. Oikos), but essentially we use the differences between food categories to weight the use of each type of food during the estimation of diet breadth. Thus, a species that consumes seeds and vertebrates (which exhibit different composition) exhibits a broader diet than another that feeds on seeds and fruits (explained in lines 122-130 SI). We nonetheless note that unweighted measures of niche breadth lead to similar conclusions (in other cases this is not true, however; see De Cáceres et al. 2011).

In addition, note that the ref. 22 is not the only reference used; the other references mistakenly appeared two lines below (refs. 24-38), we apologize for that. We have now fixed this and we also clearly explain which references were used (SI lines 132-134) for each food type (at least 2 for each food type were used).

- SI lines 156-65: clear definitions of each category of development and social mating system must be given. Across text, SI, and tables the wording used for each category and variable keep changing – please be consistent to help readers find the information more easily (e.g. social mating system in the text of the SI becomes pair-bonding in the tables).

We now better define each category, with specific examples, in the SI methods (lines 167-176). In addition, we make sure to use the same terminology everywhere to be consistent (We use the terms "social mating system" and "coloniality").

- SI line 191-5: CI for each possible transition must be reported

We agree and we now include them in Figure 3b.

- Figure S8: phylogenetic ‘correlation’ is inaccurate definition here – I suppose you mean ‘regression’ from which you computed residuals

Yes, we fixed it.

- Fig S9a: I suppose these are percentages; please clarify. If percentages, ‘carrion’ does not sum up to 100.

These are percentages. Yes, there was a typing error with Carrion, now is fixed.

- Table S6: What are the numbers for each variable in this table? Beta estimates? P-values? T-values? Why does mating system have no numbers? How are the df computed?

The numbers are the beta estimates for each factor in the model. When the variable is categorical with more than two categories (e.g. mating system), the slope is replaced by a plus sign (which just mean that the variable is included in the model). The degrees of freedom (df) are computed as the number of factors in the model (but note that mating system adds 2 df because it is a categorical variable with three levels).

- Table S7: how is the phylogenetic ANOVA carried out? Do you mean a PGLS model with a discrete predictor variable or Garland’s phylogenetic ANOVA using randomization? How can (b) represent p-values? Also for (b) there is an extra number for Resident high lat.

In the previous version, we used Garland's ANOVA because the predictors were all categorical. However, in the revised version we use a PGLS with discrete predictor, so all the models are now estimated with the same procedure.

- Tables S13-14: unclear what the last 2 columns in this table report.

This is the frequency at which we found higher values for resident from higher latitudes compared to other latitudes in one case and the frequency of higher values for short-distance migrants compared to long-distance migrants, in the other case. The reason of presenting two columns is to make sure that the comparisons between brain optima are consistent in each of the trees analysed and are not due to extreme values in few trees. We clarified this in the corresponding tables (Tables S16-S17).

Reviewer #2 (Remarks to the Author):

I appreciate that the authors have taken the time to consider and address the reviewers comments from the previous draft. I still think this paper represents an important contribution.

We thank the reviewer for the nice comment and for the previous suggestions and criticisms, which have helped us to produce a stronger work.

However, the model selection approach is still hugely ungainly and difficult to interpret. I cannot understand why, with such a large sample size, they cannot do a global model and evaluate the relative contribution of each factor. As it is, there is a plethora of different tables and models and approaches (i.e. IC versus hypothesis testing). I am less convinced by the results than I was in the previous version. There are clearly ecological and behavioural variables that are associated with brain size, but the current version does not make it straightforward to assess relative contribution.

The reason we present different models for each possible confound is to optimize sample size, as some confounding factors were not available for all the studied species. We note that our main goal with these models is to demonstrate that our results are robust to the effect of potentially confounding variables, so we feel that our approach is correct. Our model selection approach tests all possible combinations of variables and clearly shows that the full model does not explain additional variation in the response variables compared with more simplified models. However, as suggested by the referee, we now present in the appendix the full model with all the variables (Table S7), where environmental variation still significantly affects brain size. Note that some confounding factors that were significant in previous models are not anymore significant in this full model, possibly because of the reduced sample size (N=253). Therefore, we think we have to be cautious to assess relative contribution of each factor based on this reduced dataset, although we note that environmental variation has the most important effect on brain size based on both this full model and the model selection. We now explain all this in the MS, also trying to make all the alternative possibilities easier to follow (lines 177-184).

The addition of PCA is reasonable, but that there is no difference (and very little variation between low and mid-latitude environments (figure S1e) makes me question just exactly what these PCs are measuring (there are highly seasonal temperate and tropical environments, but this does not appear to be captured by these data).

Even though it is true that seasonality occurs also in low and mid latitudes, the PCA suggests that high latitudes retains more variation (see also Figure 1), as it captures most seasonal and among-years variation (PC1). This is not an unexpected result: since the axes of a PCA are orthogonal, little variation is left for the second axis. Still, there is some environmental variation independent of high-latitudes and the nice thing of the new results is that this variation is also associated with relative brain size.

I would like the authors to revisit the presentation of their results to make the alternative models easier to follow. Also, I do think there needs to be a more explicit discussion of causality and what we can infer from these results (or not)- as it is not possible to determine easily whether large brained lineages can colonise northern climates or whether large brain size is selected for in high latitudes.

Yes, we agree. In this revised version we make clearer that we cannot infer causality from our results. We do so by explicitly stating in the discussion that our approach does not reveal whether species evolved larger brains when they invaded more seasonal regions or instead their ancestors already possessed larger brains when those regions were colonized (lines 242-252 in the MS). We feel that the current version is more balanced in discussing the implications and limitations of our findings to understand the evolution of large brains.

REVIEWERS' COMMENTS:

Reviewer #1 (Remarks to the Author):

Dear authors,

I have enjoyed reading your revised ms; this is very well written and addresses all the points previously raised satisfactorily. Overall, I think this version of the ms has greatly improved and I believe that it will be of interest to the general readership of Nature Communications.

Reviewer #2 (Remarks to the Author):

In this latest revision, the authors make some effort to address some of the outstanding issues the reviewers have with the manuscript. Unfortunately, I am unable to access previous versions of the manuscript or the previous response to reviewers' comments.

I would have liked to view this information, because one of my comments in the original version was that mating system is a particularly poor way to characterise bird sociality. Nearly all passerines (the most speciose group) are socially monogamous, so variation in this trait will all occur at deep phylogenetic nodes (whereas there will be more meaningful variation in ecological traits). However, this criticism was not incorporated in reanalyses- so it is not hugely surprising that mating system is not informative. However, the authors stick with this not very informative trait. There are several papers in the literature that suggest that foraging party structure (rather than group size or mating system) is associated with relative brain size. I am fairly confident that the authors are aware of these papers, but still chose to not address this issue.

I have also repeatedly criticised the poor presentation of results (with a bazillion different tables of various combinations of ecological, behavioural and life history traits). The authors seem to insist on this- and presenting a strange combination of models in the main text (why present models that only incorporate the ecological variables that are then condensed).

For some reason that is completely beyond me, they do not want to present global models in the main text, even though the most 'global' model (which isn't) presented in the SI seems to support their case. I also find it odd that they have chosen to do a PCA on three variables, which condenses to two (really, what is the point)?

I find these issues are real shame. I still think the paper has a lot of potential, but the issues that I had with the first and second version remain. I am not quite sure what the point of peer review is when constructive advice is totally ignored. I sincerely like the topic and mostly like the approach. However, I think the statistical presentation and the characterisation of behavioural variables are flawed, and remain so after two revisions.

There are also numerous typos in the text that should be addressed (even if the substantive comments are not).

Responses to reviewers:

Comments from reviewer #1

I have enjoyed reading your revised ms; this is very well written and addresses all the points previously raised satisfactorily. Overall, I think this version of the ms has greatly improved and I believe that it will be of interest to the general readership of Nature Communications.

We sincerely appreciate all the effort the reviewer has put into our work. His/her constructive comments have helped us to improve the MS in several important ways, and we are very grateful for that.

Comments from reviewer #2

We also appreciate all the effort the reviewer has put into our work. However, we were surprised with his/her tone and latest interpretation that we might not have been fully following all her/his recommendations. Below we explain how we addressed previous concerns and we have decided to make further effort to ensure all results are crystal clear to every reader.

In this latest revision, the authors make some effort to address some of the outstanding issues the reviewers have with the manuscript. Unfortunately, I am unable to access previous versions of the manuscript or the previous response to reviewers' comments. I would have liked to view this information, because one of my comments in the original version was that mating system is a particularly poor way to characterise bird sociality. Nearly all passerines (the most speciose group) are socially monogamous, so variation in this trait will all occur at deep phylogenetic nodes (whereas there will be more meaningful variation in ecological traits).

We agree that social monogamy is not the best way to characterize sociality but it has been used in previous work on the social intelligence hypothesis (Dunbar & Shultz 2007 Science). Even when it does not reflect sociality, it has been shown to be correlated with brain size and hence could be a potential confounding factor. This is why we included it, but we acknowledge our paper is not an explicit test of sociality hypotheses.

- However, this criticism was not incorporated in reanalyses- so it is not hugely surprising that mating system is not informative. However, the authors stick with this not very informative trait.

In the first revision, the reviewer suggested to incorporate other factors apart from mating system such as "pair, bonded groups, aggregations, colony nesters or similar". We followed the advice by incorporating colonial breeding. We didn't use other social measures because they were not available for many species in published papers. In the second review, reviewer #2 did not comment anything about the need of using additional metrics, so it is unclear why now she/he raises this concern, as we interpreted the reviewer was satisfied with how we addressed this.

- There are several papers in the literature that suggest that foraging party structure (rather than group size or mating system) is associated with relative brain size. I am fairly confident that the authors are aware of these papers, but still chose to not address this issue.

Without clearly specifying the metric and the associated papers, our guess is that the reviewer is talking about Shultz & Dunbar 2010 Biol J Linn Soc, who found that social foraging structure could play a role in brain evolution. We did not use this metric in our analysis in the second revision because, as we said in our previous response, information is available for a very limited number of species (only 54 species out of 1,200 of our database were included in Shultz & Dunbar paper). In any case, we were aware of this possible confounding factor and we mentioned the paper in the previous version "other environmental factors and constraints may also influence brain size evolution (...Shultz & Dunbar 2010)". This time, as suggested by the editor, we do a last effort to include this factor. We use the data from the 54 species on Shultz & Dunbar 2010 paper and we additionally collected information on social foraging from the Handbook of Birds of the World for additional 530 species (302 residents and 228 migrants). This allows us to show that considering this measure of sociality does not alter our conclusions regarding the relation between environmental variation and brain size. This new result has been added in Table S6. We also repeated the full model (Table S7) and the model selection (Table S8) including this social foraging factor. Note however, that we still mention in the first paragraph of the discussion that the factors we use might not have captured all aspects of sociality and thus other social factors might play a role in brain size evolution.

- I have also repeatedly criticised the poor presentation of results (with a bazillion different tables of various combinations of ecological, behavioural and life history traits). The authors seem to insist on this- and presenting a strange combination of models in the main text (why present models that only incorporate the ecological variables that are then condensed).

We think that a “poor presentation of results” is a too strong criticism considering that the only arguments are the “excessive” number of tables and the need to put the full model in the main text. We already have addressed both criticisms in previous responses. The reason we present different models for each possible confound is to optimize sample size, as some confounding factors were not available for all the studied species. We note that our main goal by including these models was to test whether our results were robust to the effect of potentially confounding variables (and they are). So, in our humble opinion, this approach is correct. Our model selection approach (Table S8) tests all possible combinations of variables and clearly shows that the full model (Table S7) does not explain additional variation in the response variables as compared with more simplified models (Tables S3-S6). We agree that there are a lot of tables. But we kindly disagree that these are unnecessary and make the story too difficult to understand. The tables are in the supplementary material and do not need to be consulted to follow the main flow of the article.

- For some reason that is completely beyond me, they do not want to present global models in the main text, even though the most ‘global’ model (which isn’t) presented in the SI seems to support their case.

Again, this comment seems to imply that material in the supplementary material is little important. We kindly disagree. The only reason why we do not include this in the main text is that the table is big and we think that big tables are not so useful to communicate the main results. Having said this, to further fulfill reviewer #2 recommendations we decide to explicitly present a figure with the variable weights of the model selection (fig. 1) in the main text and we could alternatively move also the global model (Table S7) to the main text if the editor agrees with the reviewer that this is necessary.

- I also find it odd that they have chosen to do a PCA on three variables, which condenses to two (really, what is the point)?

As we explain in the text, the purpose of the PCA is to create orthogonal axes and test whether they independently affect brain size. This analysis was actually a response to a comment by reviewer #1, and its relevance is that it shows that environmental variation matters for brain size evolution regardless their nature.

- I find these issues are real shame. I still think the paper has a lot of potential, but the issues that I had with the first and second version remain.

In our humble opinion, a careful review of our previous responses to reviewers will show reviewer #2 that we indeed followed these previous suggestions, as we have explained in this response letter.

- I am not quite sure what the point of peer review is when constructive advice is totally ignored.

As we have mentioned above, we do not think this is accurate. Throughout the review process of this article we have taken very seriously all the comments of the reviewers, as even in cases when we disagree they highlight deficits in our explanations. Indeed, reviewer 1#, who made numerous -but constructive- criticisms in early versions, agrees that we dealt with all important concerns conveniently.